# Relationship Between Noninvasive Doppler-Derived Coronary Flow Reserve Measured by Transthoracic Echocardiography and Angiography Thermodilution-Measured Coronary Flow Reserve and the Index of Microcirculatory Resistance in Patients with Non-Obstructive Coronary Arteries

**DOI:** 10.3390/biomedicines13020466

**Published:** 2025-02-14

**Authors:** Milenko Čanković, Aleksandra Milovančev, Snežana Tadić, Maja Stefanović, Milovan Petrović, Mila Kovačević, Igor Tomas, Dragana Dabović, Vladimir Ivanović, Aleksandra Ilić, Anastazija Stojšić-Milosavljević, Snežana Stojšić, Nikola Komazec, Bojan Mihajlović, Igor Ivanov

**Affiliations:** 1Faculty of Medicine, University of Novi Sad, 21000 Novi Sad, Serbia; aleksandra.milovancev@mf.uns.ac.rs (A.M.); snezana.tadic@mf.uns.ac.rs (S.T.); maja.stefanovic@mf.uns.ac.rs (M.S.); milovan.petrovic@mf.uns.ac.rs (M.P.); mila.kovacevic@mf.uns.ac.rs (M.K.); dragana.dabovic@mf.uns.ac.rs (D.D.); vladimir.ivanovic@mf.uns.ac.rs (V.I.); aleksandra.ilic@mf.uns.ac.rs (A.I.); anastazija.stojsic@mf.uns.ac.rs (A.S.-M.); nikola.komazec.103@gmail.com (N.K.); bojan.mihajlovic@mf.uns.ac.rs (B.M.); igor.ivanov@mf.uns.ac.rs (I.I.); 2Institute for Cardiovascular Diseases of Vojvodina, 21204 Sremska Kamenica, Serbia; alkaliana@gmail.com (I.T.); snezana.stojsic@ikvbv.ns.ac.rs (S.S.)

**Keywords:** coronary flow reserve, index of microvascular resistance, transthoracic Doppler echocardiography, coronary microvascular dysfunction, microvascular angina

## Abstract

**Background/Objectives**: Coronary microvascular dysfunction (CMD) is emerging as a critical factor in patients presenting with anginal symptoms without obstructive coronary artery disease (CAD). This study aims to investigate the relationship between invasive measurements of coronary flow reserve (CFR) and the index of microcirculatory resistance (IMR) using thermodilution techniques, compared to non-invasive assessments of CFR with transthoracic Doppler echocardiography (TDE). **Methods**: In this observational prospective cross-sectional study, a total of 49 patients, clinically characterized as having angina with no obstructive CAD (ANOCA) or ischemia with no obstructive CAD (INOCA), underwent both TDE and invasive coronary angiography (ICA) followed by thermodilution assessment of CFR and IMR. **Results**: It was found that there is a statistically significant negative correlation between both non-invasive and invasive CFR measurements and IMR. Specifically, a negative moderate correlation was observed between non-invasive CFR and IMR (r_s_ = −0.477, *p* < 0.01), as well as a high negative correlation between invasive CFR and IMR (r = −0.541, *p* < 0.01). Receiver operating characteristic (ROC) analysis indicated that both non-invasive and invasive CFRs are effective predictors of CMD, defined as IMR > 25. **Conclusions**: Both noninvasive and invasive CFR measurements are significant independent predictors of CMD. Our results indicate that noninvasive TDE CFR can be a reliable tool for assessing CMD in patients with ANOCA, potentially facilitating earlier diagnosis and management strategies for this patient population.

## 1. Introduction

Coronary microvascular dysfunction (CMD) has increasingly been linked to impaired quality of life and adverse outcomes in recent years. Despite this growing body of evidence, CMD remains significantly underdiagnosed and undertreated in the clinical setting [1]. Therefore, interest in assessing CMD through different noninvasive imaging (transthoracic Doppler echocardiography (TDE), positron emission tomography (PET), and cardiac magnetic resonance imaging (CMR)) and invasive hemodynamic diagnostic methods (intracoronary temperature-pressure wire/intracoronary Doppler flow-pressure wire testing of coronary flow reserve (CFR) and/or microvascular resistance index (IMR), and intracoronary provocation testing) in these patients with microvascular angina (MVA) has significantly increased [2,3].

Based on clinical and diagnostic tests, we differentiate three clinical phenotypes of CMD: angina with no obstructive coronary arteries (ANOCA), ischemia with no obstructive coronary artery disease (INOCA), and, in some cases, patients may present with myocardial infarction despite having no obstructive coronary artery disease (MINOCA) [4].

CMD also plays a role in several other cardiovascular diseases, such as obstructive coronary artery disease (CAD), non-ischemic cardiomyopathies, Takotsubo syndrome, and heart failure (HF), particularly in patients with preserved ejection fraction [5].The latest guidelines from the European Society of Cardiology endorse invasive coronary functional testing as the primary diagnostic approach (Class of Recommendation Ib) for patients with chronic coronary syndrome who continue to experience symptoms despite medical therapy, particularly in cases of suspected ANOCA or INOCA. Additionally, while non-invasive testing is also recognized as a viable option, it is assigned a lower recommendation class (IIb), mainly due to a lack of substantial evidence [6].

Transthoracic Doppler echocardiography (TDE) is a non-invasive method used to assess CFR in the epicardial coronary arteries [7]. Both noninvasively measured and invasively measured CFR provide an assessment of the blood flow in both the epicardial coronary arteries and the heart microcirculation and can be used to diagnose CMD in patients without obstructive epicardial coronary artery disease [8]. In contrast to CFR, IMR is not influenced by epicardial vascular function and is specifically determined by microcirculation resistance. An IMR greater than 25 is considered abnormal and aligns with a diagnosis of CMD [9].

Therefore, addressing the knowledge gap in assessing the relationship between IMR and TDE assessment of CFR on the left descending artery (LAD) and invasive evaluation of CFR could be fundamental in understanding the correlation between non-invasive and invasive methods. This could significantly improve the diagnostic accuracy of these diagnostic tools and patient management strategies. Furthermore, exploring this relationship may help identify patients at risk of microvascular dysfunction, often overlooked in the standard diagnostic pathway.

We aimed to assess the relationship between IMR and noninvasive Doppler-derived CFR measured by transthoracic echocardiography and angiography thermodilution-measured CFR in patients who clinically presented as ANOCA and INOCA.

## 2. Materials and Methods

In this observational prospective cross-sectional study, we aimed to include patients with symptoms of anginal chest pain without obstructive CAD on computed tomography coronary angiography (CCTA) to assess CMD or positive non-invasive ischemia testing accompanied with TDE suggesting CMD. Patients with symptoms of angina without obstructive coronary artery disease are defined as ANOCA, while patients with positive tests on ischemia are defined as INOCA [3,10].

One hundred fifteen consecutive patients admitted to a tertiary care center with signs and symptoms of angina were screened for inclusion in the study. The exclusion criteria for the study were:Obstructive CAD in at least one coronary artery characterized by stenosis >90% or chronic total occlusion (CTO) on CCTA or invasive coronary angiography (ICA).Fractional flow reserve (FFR) indicating significant obstructive CAD in patients with coronary artery stenosis 50–90% on ICA.Inability to perform TDE due to the poor echocardiographic window or some other reason.Inability to position the FFR wire in the distal LAD due to the tortuosity.Registering the vasospasm on the ICA before or after positioning the FFR wire in distal LAD.Patients with prior myocardial infarction or MINOCA.Patients with elevated troponin.

Due to the obstructive CAD on CTCA (lesions ≥ 90%, CTO, or multi-vessel disease), 39 patients were excluded. In total, 76 patients underwent TDE assessment and ICA. Based on single or multi-vessel disease, CTO, or abnormal FFR indicating significant obstructive CAD, 27 patients were excluded from the study and subsequently underwent revascularization. In the end, 49 patients were included in the study and underwent invasive assessment of CMD with thermodilution measurement of CFR and IMR. All included patients underwent a medical history assessment, which included evaluating risk factors and comorbidities and examining clinical findings. Based on their cardiologist’s decision, the patients received standard treatment for angina, including calcium antagonists, beta-blockers, trimetazidine, nitrates, ranolazine, and additional therapies tailored to individual risk factors and comorbidities.

TDE and CFR measurements were performed using a commercially available machine (Vivid XD Clear; GE Healthcare, Milwaukee, WI) with an M5S probe. Heart rate and blood pressure were recorded at the same time as echocardiography. Non-invasive CFR was assessed by TDE using intravenous adenosine infusion (140 mg/kg/min over 2 min) to establish the condition of maximal hyperemia. Briefly, the mid-distal part of the LAD was studied, and the artery was visualized using color Doppler flow mapping guidance in the modified parasternal view. For color Doppler echocardiography, the velocity range was defined from 12 to 19 cm/s. Blood flow velocity was measured by pulsed-wave Doppler echocardiography, using a sample volume of 3 to 4 mm, placed on the color signal in the distal LAD. The ultrasound beam direction was aligned as closely as possible with the distal LAD flow. CFR was calculated as the hyperemic to basal peak diastolic flow velocity ratio. The final values of flow velocity represented an average of three cardiac cycles. Invasive assessment of CFR and IMR was performed after TDE measurement of CFR in patients without obstructive coronary artery disease (lesion < 50% on coronary angiography) on ICA. Interventional cardiologists were blinded to the previous findings of the TDE CFR measurement.

ICA was performed by using transradial access. After diagnostic ICA, in which no obstructive coronary artery disease was registered, CMD was assessed by measuring CFR and IMR using the thermodilution technique on the CoroFlow platform (Coroventis, Cardiovascular System, Coroventis Research, Uppsala, Sweden). A guiding catheter was positioned in the left coronary area (LCA), followed by the insertion of the wire with a pressure and temperature sensor (Pressure wire X, Abbott Vascular, Santa Clara, CA, USA) in the distal segment of LAD, at least 60 mm from the tip of the guiding catheter. Using the CoroFlow platform (Coroventis, Cardiovascular System, Coroventis Research, Uppsala, Sweden), mean transit time was calculated in LAD by using the thermodilution technique with the application of 3 mL of room-temperature saline at least three times. After excluding outliers and acquiring adequate mean transit time, maximal hyperemia was induced by the intracoronary application of 15 mg of papaverine. During the maximal hyperemia, mean transit time measurement was repeated with the application of 3 mL of room-temperature saline at least three times. Again, after excluding outliers, mean transit time in maximal hyperemia was acquired. After careful assessment of the curves on the CoroFlow platform (Coroventis), CFR and IMR were acquired, as well as the non-hyperemic pressure ratio (NHPR) of the pressure in the aorta and at the pressure wire in the distal LAD (Pd/Pa ratio) and fractional flow reserve (FFR) in the condition of maximal hyperemia. [6,11]. Finally, non-invasively measured CFR_non-inv_ by TDE was compared with invasively measured CFR_inv_, IMR, Pd/Pa, and FFR.

### Statistics

The normality of data was assessed using the Shapiro–Wilk test. Based on distribution, continuous variables are presented as means with standard deviations (SD) or medians with interquartile ranges between the 25th (Q1) and the 75th (Q3) percentiles for continuous data. Categorical variables are presented as absolute numbers and percentages. Differences between groups were assessed using the Kruskal–Walis test for not normally distributed continuous data. The correlation between different variables (indexes) was analyzed by calculating the Spearman correlation coefficient (r_s_) or Pearson correlation coefficient (r). Additionally, linear regression models were estimated to evaluate the relationship between different hemodynamic indexes and Doppler echocardiography-measured CFR. Linear regression analysis was performed to identify independent predictors of IMR > 25. The predictive quality of the variables on IMR > 25 was evaluated using receiver operating characteristic (ROC) curves. The area under the curve (AUC) was obtained to assess its diagnostic performance, and an AUC comparison was performed using DeLong’s method. A *p*-value of 0.05 was considered to indicate statistical significance. Statistical software Stata (Stata 18.0 StataCorp 4905 Lakeway Drive, College Station 77845, TX, USA, licensed to the University of Novi Sad) was used for all calculations.

## 3. Results

The study included 49 patients. The median age was 63 (57, 69), with a higher prevalence of females. There were no significant differences between the median ages of males and females. A high prevalence of risk factors was observed. The most prevalent risk factor was hypertension, followed by hyperlipidemia, family history of cardiovascular disease (CVD), and smoking. One-third (32.6%) had two or more comorbidities, such as hypertension combined with hyperlipidemia. The majority of patients, 95.9%, had a preserved EF of ≥50%; 2 patients (4.1%) had an EF between 40 and 50%. Table 1 presents baseline patient characteristics.

In Table 2, echocardiographic Doppler CFR and measurements derived from ICA were presented. The median non-invasive CFR measured by TDE was lower than the mean CFR invasively measured. Non-invasive CFR was ≤2 in 30 (61.2%) patients, while 26 (53.1%) patients had IMR > 25.

### 3.1. Correlation of Doppler Echocardiography-Measured CFR and Invasive Coronary Angiography-Measured Parameters

A statistically significant negative moderate correlation was observed between noninvasive CFR_non-inv_ and IMR, r_s_ = −0.477, *p* < 0.01 (Figure 1), as well as a significant positive high correlation between noninvasive and invasive CFR_inv_ r_s_ = 0.504, *p* < 0.01. There is a low statistically not significant correlation between noninvasive CFR_non-inv_ and NHPR (Pd/Pa ratio) in basal conditions (r_s_ = 0.159, *p* = 0.27) as well as between noninvasive CFR_non-inv_ and FFR (r_s_ = 0.056, *p* = 0.69).

### 3.2. Correlation of Invasive Coronary Angiography-Measured CFR and Angiographic Parameters

There is a statistically significant high negative correlation between invasive CFR_inv_ and IMR, r = −0.541, *p* < 0.01 (Figure 2). There is no statistically significant correlation between invasive CFR_inv_ and NHPR (Pd/Pa), r = −0.113, *p* = 0.437, as well as FFR r = −0.059, *p* = 0.68.

There is no statistically significant correlation between IMR and NHPR (Pd/Pa) (r = 0.116, *p* = 0.424) or between IMR and FFR (r = 0.262, *p* = 0.068).

### 3.3. Linear Regression

There is a significant association between noninvasive CFR_non-inv_ and invasive CFR_inv_; for one unit increase in invasive CFR_inv_, noninvasive CFR increases by 0.191, 95% CI −0.059; −0.322, *p* < 0.01, as well as a significant association between IMR and CFR_non-inv._ For one unit increase in IMR, noninvasive CFR_non-inv_ decreases by 0.016, 95% CI −0.028; −0.005, *p* < 0.01 (Table 3).

There is a significant association between IMR and invasive CFR. An increase in one unit of IMR leads to an estimated decrease of 0.048 in invasive CFR, *p* < 0.01 (Table 4).

In multivariable linear regression, invasive CFR_inv_ was found to be the strongest predictor of IMR > 25 (Table 5).

### 3.4. ROC Curves

Both invasive and non-invasive CFR are good predictors of CMD (defined as IMR > 25). The ROC curve showed that the non-invasive CFR_non-inv_ is a good predictor for IMR > 25 with a corresponding AUC of 0.8227 (95% CI 0.697–0.948) with a high sensitivity of 92.31% and specificity of 73.91%, a positive predictive value of 80%, and a negative predictive value of 89.47%. The empirical optimal cut-off was found to be 1.63. Similarly, the AUC for the ROC curve for invasive CFR_inv_ was 0.836 (95% CI 0.725–0.946) with 76.9% sensitivity, 73.91% specificity, a PPV of 76.92%, and an NPV of 73.91% (Figure 3). The empirical optimal cut-off was found to be 3.5. There is no difference between these two models in predicting IMR > 25, *p* = 0.8514.

## 4. Discussion

In our study, we found a statistically significant high negative correlation between CFR_inv_ and IMR and a statistically significant negative moderate correlation of noninvasive CFR_non-inv_ and IMR in patients who clinically presented as ANOCA and INOCA. Both linear regression models with CFR_non-inv_ and CFR_inv_ were good predictors of CMD with high sensitivity and specificity. CFR_non-inv_ correlates well with CFR_inv_. These results suggest that TDE is a viable alternative to invasive thermodilution measurements for assessing CFR.

These findings highlight the importance of noninvasively measured CFR in establishing the diagnosis of CMD. With a high sensitivity of 92.31% and specificity of 73.91% at an empirical cut-off of 1.6, determining a diagnosis of CMD could be timely and facilitated as noninvasive techniques are more easily accessible. Patients with low noninvasively measured CFR should be referred more promptly to ICA for diagnosis confirmation and treatment initiation. Furthermore, TDE-derived high values of CFR_non-inv_, along with non-obstructive coronary arteries on CCTA, are less likely to have CMD. However, this method cannot exclude vasospastic angina. Additional data on this finding and its applications are needed. Randomized controlled trials comparing major adverse cardiac events in patients diagnosed with CMD, both non-invasively and invasively, would provide crucial information. The relationship between CFR and IMR is complex in patients with ANOCA, and it is dependent on the type of CMD. In patients with structural CMD, we can expect to find low CFR < 2.5 and high IMR > 25; however, patients with functional CMD can have low CFR < 2.5 and normal IMR < 25. It is important to emphasize that the risk of major adverse cardiac events (MACE) is the same between these two groups [12].

Around 70% of the patients with ANOCA who are being referred to coronary angiography do not have obstructive coronary arteries, yet around 25% of them have detectable ischemia on non-invasive tests [6]. This is a large number of patients who need additional testing. Current guidelines recommend invasive techniques for both the assessment of CMD and Vasospastic angina (VSA), which has made these techniques standard practice in most catheterization laboratories. However, the global implementation of the guidelines in real-life practice is still unknown. Furthermore, local practices in different catheterization laboratories in the selection of patients who are being referred for assessment of CMD may differ, and this is currently being studied by several prospective registries [13].

TDE is a non-invasive, safe, inexpensive, broadly available technique that can be used with high sensitivity and specificity for the measurement of CFR. In cases when coronary anatomy is unknown, TDE CFR < 2 can imply a significant lesion on LAD, or in cases without obstructive coronary artery disease, it can imply CMD [7,14]. In the era in which CCTA is being used in daily practice to assess coronary anatomy in patients with symptoms of angina or its equivalent, also with or without signs of ischemia, adding TDE CFR assessment in patients without obstructive coronary artery disease can be used to diagnose CMD. However, the correlation of non-invasive CFR assessment with TDE and invasive evaluation of CFR and IMR with the thermodilution technique is unknown.

To our knowledge, this study is the first to correlate non-invasive TDE CFR and invasive measurement of CFR and IMR with the thermodilution technique.

Also, we found that non-invasive CFR, invasive CFR,_,_ and IMR do not have a significant correlation with NHPR and FFR. The data about the CFR correlation with FFR in patients with non-obstructive coronary arteries are lacking. Still, the relationship between FFR and CFR is complex and dependent on different factors such as the severity of epicardial stenosis and the presence of CMD [15,16]. However, recently Galante et al. in their study showed that globally there is no correlation between the IMR and the FFR in patients with ANOCA [17].

When it comes to the major adverse cardiac events (MACE) related to the values of CFR and IMR, according to the data from Boerhout et al., abnormal CFR was related to the increased risk of the MACE during the 5-year follow-up period regardless of the IMR values. However, in the same study, the IMR value > 25 was not related to the increased risk of the MACE during the same follow-up period [12]. Furthermore, low CFR (<2) determined by TDE has been shown to be an independent predictor of the MACE during the median follow-up period of 4.5 years, but in patients with unknown coronary anatomy [18]. Yet, TDE measurement of CFR in patients with ANOCA has unknown predictability of MACE in patients with low CFR, and future trials are needed. Even though it is a safe, non-invasive, and not expensive technique, TDE measurement of CFR has some limitations, such as a poor acoustic window or inability to locate the LAD for the measurements [5].

Our study had some limitations that could potentially affect results. Although we found a strong correlation, the findings should be interpreted with caution due to the limitation posed by the small sample size. Studies with a higher number of patients are needed to confirm our results. The study was conducted at a single center, which may limit the generalizability of the findings to broader populations. The demographic and clinical characteristics of the study population treated in a tertiary care center may not reflect the diversity found in the general population, potentially affecting the applicability of results. Patients were not examined for VSA since acetylcholine is not available on the market. Also, adenosine was the hyperemic agent for the TDE measurement of the CFR, while papaverine was used to induce maximal hyperemia for invasive assessment. Differences in measurement techniques for CFR and IMR could introduce variability in results and affect comparisons between non-invasive and invasive assessments. These limitations highlight areas where further investigation is needed.

## 5. Conclusions

In this study, we showed that the noninvasive TDE measurement of CFR strongly correlates with invasively measured CFR and IMR by the thermodilution technique. Therefore, it may be used to facilitate and promptly diagnose CMD in patients with non-obstructive coronary artery disease.

## Figures and Tables

**Figure 1 biomedicines-13-00466-f001:**
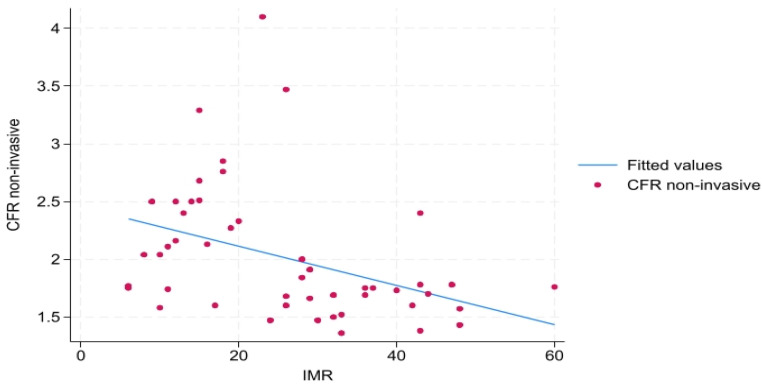
Scatter plot of Doppler echocardiography-measured CFR and invasive coronary angiography-measured IMR. Legend: IMR—index of microvascular resistance, CFR non-invasive—transthoracic Doppler echocardiography-derived coronary flow reserve.

**Figure 2 biomedicines-13-00466-f002:**
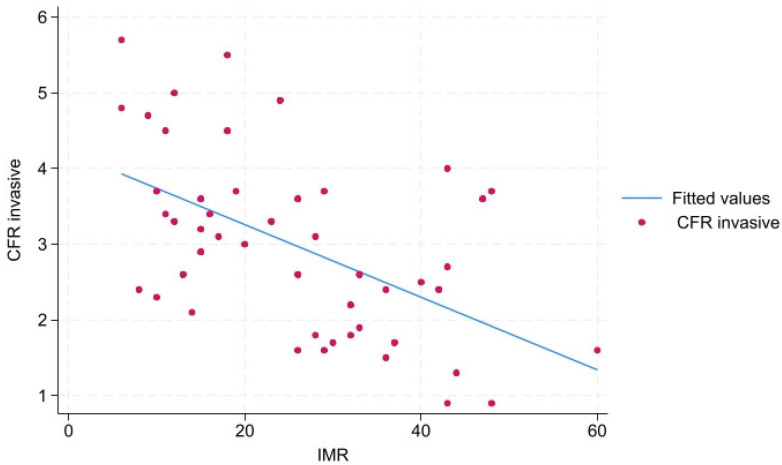
Scatter plot invasive coronary angiography-measured CRF and IMR. Legend: IMR—index of microvascular resistance, CFR_invasive_—invasive angiography-derived coronary flow reserve.

**Figure 3 biomedicines-13-00466-f003:**
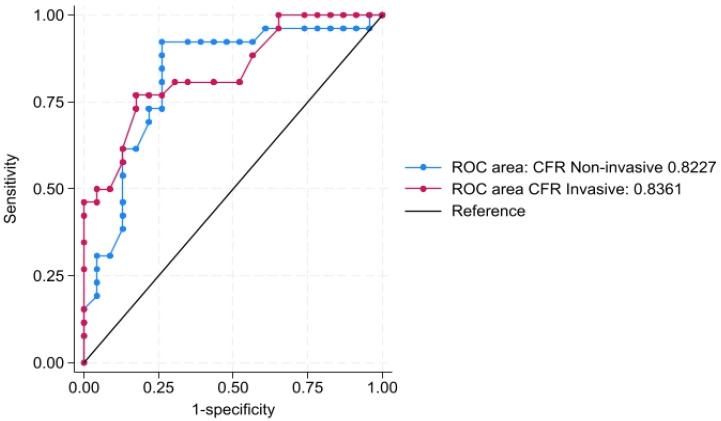
ROC curves, including noninvasive and invasive coronary flow reserve (CFR) for predicting IMR > 25. Legend: ROC—receiver-operating characteristic curve, CFR_invasive_—invasive angiography-derived coronary flow reserve, CFR_non-invasive_—transthoracic Doppler echocardiography-derived coronary flow reserve.

**Table 1 biomedicines-13-00466-t001:** Baseline patient characteristics (N = 49 for all).

Parameter	N (%) Median (Q1, Q3) Mean (SD)
Gender (F)	30 (61.2%)
Age (years)	63 (57, 69)
BMI (kg/m^2^)	26.9 (3.7)
Family history of CVD	27 (55.1%)
History of hypertension	41 (83.7%)
Smoking history	23 (46.9%)
History of hyperlipidemia	30 (61.2%)
History of diabetes	12 (24.5%)
History of obesity	10 (20.4%)
EF (%)	60 (60, 63)

Legend: BMI—Body mass index, CVD—Cardiovascular disease, EF—Ejection fraction, Female—F, Q1—25th and Q3—75th percentile.

**Table 2 biomedicines-13-00466-t002:** Noninvasive Doppler-derived coronary flow reserve, measured by transthoracic echocardiography and invasive coronary angiography-measured parameters (N = 49 for all).

Parameter	N (%) Median (Q1, Q3) Mean (SD)	95% CI
CFR Noninvasive	1.8 (1.6; 2.3)	1.843; 2.175
CFR Invasive	2.9 (1.2)	2.630; 3.312
FFR basal	0.92 (0.9; 0.94)	0.916; 0.934
FFR (Papaverin)	0.87 (0.05)	0.852; 0.881
IMR	26 (13.4)	22.146; 29.853

Legend: CFR—coronary flow reserve, FFR—fractional flow reserve, IMR—index of microcirculatory resistance, CI—confidence interval, SD—standard deviation, Q1—25th percentile, and Q3—75th percentile.

**Table 3 biomedicines-13-00466-t003:** Coefficients and 95% confidence intervals (CI) of the linear regression models, including noninvasive coronary flow reserve (CFR) with different invasive coronary angiography-measured parameters.

CFR_non-inv_	Coefficient	*p*	95% CI
CFR_inv_	0.191	<0.01	−0.059; −0.322
IMR	−0.016	<0.01	−0.028; −0.005
NHPR (Pd/Pa)	3.165	0.260	−2.425; 8.756
FFR	−0.591	0.730	−4.023; 2.841

Legend: CFR_non-inv_—transthoracic Doppler echocardiography-derived coronary flow reserve, CFR_inv_—invasive angiography-derived coronary flow reserve, IMR—index of microcirculatory resistance, NHPR (Pd/Pa)—non-hyperemic pressure ratio (distal pressure/aortic pressure), FFR—fractional flow reserve.

**Table 4 biomedicines-13-00466-t004:** Coefficients and 95% confidence intervals (CI) of the linear regression models, including invasive coronary flow reserve (CFR) with different invasive coronary angiography-measured parameters.

CFR_inv_	Coefficient	*p*	95% CI
IMR	−0.048	<0.01	−0.069; −0.026
NHPR (Pd/Pa)	4.503	0.437	−7.059; 16.066
FFR	−1.427	0.685	−8.471; 5.616

Legend: CFR_inv_—invasive angiography-derived coronary flow reserve, IMR—index of microcirculatory resistance, NHPR (Pd/Pa)—non-hyperemic pressure ratio (distal pressure/aortic pressure), FFR-fractional flow reserve.

**Table 5 biomedicines-13-00466-t005:** Multivariable linear regression model assessing predictors of IMR > 25.

IMR > 25	Coefficient	Std. Err.	t	*p*	95% CI	
CFR_inv_	−0.205	0.051	−0.401	<0.01	−0.308	−0.102
CFR_non-inv_	−0.263	0.105	−2.510	0.02	−0.474	−0.052

Legend: IMR—index of microcirculatory resistance, CFR_inv_—invasive angiography-derived coronary flow reserve, CFR_non-inv_—transthoracic Doppler echocardiography-derived coronary flow reserve.

## Data Availability

The original contributions presented in this study are included in the article. Further inquiries can be directed to the corresponding author.

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
