# Peer review of "Relationship Between Noninvasive Doppler-Derived Coronary Flow Reserve Measured by Transthoracic Echocardiography and Angiography Thermodilution-Measured Coronary Flow Reserve and the Index of Microcirculatory Resistance in Patients with Non-Obstructive Coronary Arteries"

_biomedicines, 2025, doi:10.3390/biomedicines13020466_

Round 1
Reviewer 1 Report
Comments and Suggestions for Authors
The paper investigate relationship between invasive measurements of coronary flow reserve and IMR. A correlation was found between them which can be useful to diagnose coronary microvascular dysfunction.
I didn't found significant flaws in paper, looks fair enough for initial research Figures 1-3 looks like not perfect, but due to pdf conversion settings.
Did you try or plan to use nonlinear regression for such study?
Author Response
The paper investigate relationship between invasive measurements of coronary flow reserve and IMR. A correlation was found between them which can be useful to diagnose coronary microvascular dysfunction.
I didn't found significant flaws in paper, looks fair enough for initial research Figures 1-3 looks like not perfect, but due to pdf conversion settings.Did you try or plan to use nonlinear regression for such a study?
Answer: Thank you for your valuable feedback. As we observed linear regression with strong linear correlation, there was no need to do nonlinear regression.
Reviewer 2 Report
Comments and Suggestions for Authors
Review of the manuscript ”Relationship between noninvasive Doppler derived coronary flow reserve, measured by transthoracic echocardiography and angiography thermodilution measured coronary flow reserve and index of microcirculatory resistance in patients with non-obstructive coronary arteries” for the Biomedicines journal.
The goal of this study was to investigate the relationship between invasive measurements of coronary flow reserve (CFR) and the index of microcirculatory resistance (IMR) using thermodilution techniques, compared to non-invasive assessments of CFR with transthoracic Doppler echocardiography (TDE). Authors have shown that the TDE measurement of CFR has a good correlation with invasively measured CFR and IMR by thermodilution technique. In relation to that, it may be used in patients with non-obstructive coronary artery disease to diagnose CMD.
However, during the review of this manuscript though, some remarks and comments appeared.
Minor comments:
1. The text of the manuscript should be corrected by a native English speaker.
2. There are some spelling errors throughout the manuscript that should be carefully corrected during revision.
3. The number of patients in the study group seems to be small and the results should be confirmed in an independent patients’ group.
4. Figures 1, 2 – Description of the y axis is missing.
Comments on the Quality of English Language
The text of the manuscript should be corrected by a native English speaker.
Author Response
Thank you for your valuable feedback. We made corrections to the article based on your suggestions.
Review of the manuscript ”Relationship between noninvasive Doppler derived coronary flow reserve, measured by transthoracic echocardiography and angiography thermodilution measured coronary flow reserve and index of microcirculatory resistance in patients with non-obstructive coronary arteries” for the Biomedicines journal.
The goal of this study was to investigate the relationship between invasive measurements of coronary flow reserve (CFR) and the index of microcirculatory resistance (IMR) using thermodilution techniques, compared to non-invasive assessments of CFR with transthoracic Doppler echocardiography (TDE). Authors have shown that the TDE measurement of CFR has a good correlation with invasively measured CFR and IMR by thermodilution technique. In relation to that, it may be used in patients with non-obstructive coronary artery disease to diagnose CMD.
However, during the review of this manuscript though, some remarks and comments appeared.
Minor comments:
- The text of the manuscript should be corrected by a native English speaker.
Answer: English has been corrected.
- There are some spelling errors throughout the manuscript that should be carefully corrected during revision.
Answer: Spelling errors have been corrected.
- The number of patients in the study group seems to be small and the results should be confirmed in an independent patients’ group.
Answer: We acknowledge this comment and have included sample size as a limitation of the study. Furthermore, additional research on this topic is necessary.
- Figures 1, 2 – Description of the y axis is missing.
Answer: Figures 1 and 2 have been corrected adding the description of the y-axis.
Reviewer 3 Report
Comments and Suggestions for Authors
The article entitled "Relationship between noninvasive Doppler derived coronary flow reserve, measured by transthoracic echocardiography and angiography thermodilution measured coronary flow reserve and index of microcirculatory resistance in patients with non-obstructive coronary arteries" is interesting and compare a less invasive method for microvascular angina with ICA. However, some adjustments need to be made:
1. Which were the exclusion criteria?
2. In the result section, don't repeat the percentage in the text since they are already mentioned in the tables.
3 .Same aspect for mean +-SD and median + IQR. The results are already in the table and they are duplicate since they are written again in the text. In the manuscript, it is necessary just to present the interpretation of the results.
4 .What about medication that the patients took? It is important to mention. Also, were patients with elevated troponin levels included? If not, the authors need to extend the exclusion criteria.
5. The authors mentioned that they included patients with angina. For how long did the patients had angina? And according to the CCS classification, how many patients had Class I/II/III/IV? Can the severity of the symptoms influence the results?
6. The authors need to extent the list of the limitations. First of all, the number of the patients is relatively small. Then, CCTA can sometimes supraestimate the results or give some false positive/false negative results. Please comment.
7. In the discussion section, this paragraph "In our study, we found that there is a statistically significant high negative correlation between invasive CFRinv and IMR, as well as a statistically significant negative moderate correlation of noninvasive CFRnon-inv and IMR in patients who clinically presented as ANOCA and INOCA. Both models with CFRnon-inv and CFRinv were good predictors of CMD with high sensitivity and specificity." just repeats the results. The authors need to explain those results, which one is higher/lower and when.
8. The authors need to mention why this study is important in clinical practice (e.g. less invasive method for TDE), for which kind of patients is necessary to perform TDE rather than ICA, and some future perspectives.
Author Response
Thank you for your valuable feedback which significantly improved the manuscript. We have added some corrections to the article based on your suggestions.
The article entitled "Relationship between noninvasive Doppler derived coronary flow reserve, measured by transthoracic echocardiography and angiography thermodilution measured coronary flow reserve and index of microcirculatory resistance in patients with non-obstructive coronary arteries" is interesting and compare a less invasive method for microvascular angina with ICA. However, some adjustments need to be made:
- Which were the exclusion criteria?
Answer: We agree that this is important information, and we added additional text to the methodology.
- In the result section, don't repeat the percentage in the text since they are already mentioned in the tables.
- Same aspect for mean +-SD and median + IQR. The results are already in the table and they are duplicate since they are written again in the text. In the manuscript, it is necessary just to present the interpretation of the results.
Answer 2,3: Corrected. Now, there is no repeated text except age, which we consider important to mention in the form of the median (IQR). Different percentages are calculated and mentioned in the text and different in the tables.
4 .What about the medication that the patients took? It is important to mention. Also, were patients with elevated troponin levels included? If not, the authors need to extend the exclusion criteria.
Answer: This information has been added to the methodology section. If specific percentages are required, we can include those, but we need more time to verify the details.
Answer: We updated the exclusion criteria.
- The authors mentioned that they included patients with angina. For how long did the patients had angina? And according to the CCS classification, how many patients had Class I/II/III/IV? Can the severity of the symptoms influence the results?
Answer: This research question is very interesting, although it was not the primary aim of our research. This hypothesis (Can the severity of the symptoms influence the CMD?) can be a subject of future research. All of our patients had variable stages and duration of angina. To examine this kind of association, a higher number of patients would be necessary, as there are four classes of angina.
If you think these results are necessary for this research, we will need more time than 10 days for revision to verify and examine this.
- The authors need to extent the list of the limitations. First of all, the number of the patients is relatively small. Then, CCTA can sometimes supraestimate the results or give some false positive/false negative results. Please comment.
Answer: Thank you for this important comment. We updated the limitations section regarding sample size and other important questions.
We excluded the patients who had severe obstructive coronary artery disease (stenosis >90%) on CCTA. All of those patients underwent ICA which confirmed significant coronary artery disease.
- In the discussion section, this paragraph "In our study, we found that there is a statistically significant high negative correlation between invasive CFRinv and IMR, as well as a statistically significant negative moderate correlation of noninvasive CFRnon-inv and IMR in patients who clinically presented as ANOCA and INOCA. Both models with CFRnon-inv and CFRinv were good predictors of CMD with high sensitivity and specificity." just repeats the results. The authors need to explain those results, which one is higher/lower and when.
- The authors need to mention why this study is important in clinical practice (e.g. less invasive method for TDE), for which kind of patients is necessary to perform TDE rather than ICA, and some future perspectives.
Answer 7,8: These two initial sentences are distinct: the first highlights the results of the linear correlation, while the second evaluates the outcomes derived from linear regression. The explanations are in the results section.
A discussion section has been added with additional explanation.
In our study, we found a statistically significant high negative correlation between CFRinv and IMR and a statistically significant negative moderate correlation of noninvasive CFRnon-inv and IMR in patients who clinically presented as ANOCA and INOCA. Both linear regression models with CFRnon-inv and CFRinv were good predictors of CMD with high sensitivity and specificity. CFRnon-inv correlates well with CFRinv. These results suggest that TDE is a viable alternative to invasive thermodilution measurements for assessing CFR.
These findings highlight the importance of noninvasively measured CFR in establishing the diagnosis of CMD. With a high sensitivity of 92.31% and specificity of 73.91% at an empirical cut-off of 1.6, determining a diagnosis of CMD could be timely and facilitated as noninvasive techniques are more easily accessible. Patients with low noninvasively measured CFR should be referred more promptly to ICA for diagnosis confirmation and treatment initiation. Furthermore, TDE-derived high values of CFRnon-inv, along with non-obstructive coronary arteries on CCTA, are less likely to have CMD. However, this method cannot exclude vasospastic angina. Additional data on this finding and its applications are needed. Randomized controlled trials comparing major adverse cardiac events in patients diagnosed with CMD, both non-invasively and invasively, would provide crucial information.
Round 2
Reviewer 3 Report
Comments and Suggestions for Authors
Congratulation to the authors for making all the necessary changes. The article is now suitable for publication.